# An Exploratory Study of EFL Learners' Use of ChatGPT for Language Learning Tasks: Experience and Perceptions

## Yangyu Xiao [1] and Yuying Zhi [2,*]

1   School of Humanities and Social Science, The Chinese University of Hong Kong, Shenzhen 518172, China; shirleyxiao@cuhk.edu.cn
2   College of Foreign Languages, University of Shanghai for Science and Technology, Shanghai 200093, China
*   Correspondence: zhiyuying@usst.edu.cn

**Abstract:** ChatGPT, a general-purpose intelligent chatbot developed by OpenAI, has introduced numerous opportunities and challenges in the field of language education. With its remarkable ability to generate diverse forms of text, answer questions, and provide translations within minutes, ChatGPT has become an influential tool in the era of advanced AI technology. However, to what extent ChatGPT can be used to assist students in completing language learning tasks remains largely unexplored. Against this background, this study aimed to investigate students' experiences with ChatGPT and their perceptions of its role in language learning through a small-scale qualitative study. The data were collected through semi-structured interviews with five students at a top-tier international university in China. Students' responses revealed that ChatGPT has the potential to serve as a valuable learning partner and aid students in completing language-related tasks. Furthermore, participants exhibited critical judgment in evaluating the quality of ideas and outputs generated by ChatGPT, as well as the ability to modify prompts to maximize learning benefits. Such critical judgment offsets the potential threats to academic integrity posed by ChatGPT. Our findings contribute to the understanding of the potential of ChatGPT in language education by adding empirical evidence from students' perspectives. This study supports the idea that ChatGPT can work as an effective tool for providing students with immediate feedback and personalized learning experiences. Such findings generate implications for future pedagogical practices in the new era by providing students with personalized guidance, designing technology-embedded language support, and developing students' lifelong learning skills (e.g., autonomy and evaluative judgment) with the support of ChatGPT.

**Keywords:** ChatGPT; language learning; learners' perception; EFL learners

## 1. Introduction

The potential of technology-enhanced learning tools to support language learning has attracted significant attention from educational scholars and practitioners across various fields, particularly following the release of ChatGPT by OpenAI in November 2022. Technology provides unique opportunities for language learners to interact with the target language, offering authentic and interactive resources that facilitate the development of language skills in a more immersive and meaningful manner (Hassan Taj et al. 2017; Loncar et al. 2021). Moreover, technology enhances learners' motivation and autonomy, empowering them to assume control over their learning process (Gikas and Grant 2013; Peterson 2017). One notable technological innovation in this realm is AI-powered chatbots. Through natural language processing and machine learning techniques, these chatbots engage learners in dialogue-based interactions, providing personalized and interactive language learning experiences (Guo et al. 2022). They have the ability to adapt to learners' proficiency levels, provide real-time feedback, and foster learner autonomy and self-correction (Chen et al. 2021).

ChatGPT, developed by OpenAI, is an AI-powered chatbot that utilizes large-scale language models to generate human-like text (Bender et al. 2021). Its potential benefits for language learning are diverse, including assistance in language skill development, the provision of personalized practice materials, and support for writing, research, and problem-solving tasks (Kasneci et al. 2023). While ChatGPT seems to have become an innovative and revolutionary tool for language education, concerns have also emerged regarding the potential risks associated with its inappropriate use, such as fairness, copyright infringement, and breaches of academic integrity (Kasneci et al. 2023). Criticisms have also been raised regarding the accuracy and reliability of the information generated by ChatGPT.

Despite the growing significance of technology, including tools such as ChatGPT, in language learning, there exists a notable research gap regarding the impact and effectiveness of these tools from the learners' perspective. While previous studies have highlighted the potential benefits of AI-powered chatbots, there is a need to explore how learners perceive and engage with ChatGPT, as well as address their unique needs and challenges (Jeon 2021). This research gap calls for further investigation to bridge the knowledge divide and gain insights into optimizing the use of ChatGPT for language learning. The current study aims to address this research gap by examining learners' perceptions of the role of ChatGPT when they complete language-learning tasks. By comprehending how learners perceive the role of ChatGPT in language learning and examining their experiences, attitudes, and interactions with the technology, a thorough understanding can be obtained. This understanding is critical for optimizing the use of ChatGPT to cater to the specific needs and challenges of language learners.

## 2. Literature Review

### 2.1. The Role of ChatGPT in Technology-Enhanced Language Learning

The integration of technology in language learning has become increasingly vital in the field. Technology offers unique opportunities for language learners to engage with the target language (Hassan Taj et al. 2017). One of the key advantages lies in the access to authentic and interactive resources provided by technology (Loncar et al. 2021). Learners benefit from a wide range of authentic language materials, such as videos, online articles, and podcasts, which reflect real-life language use. This exposure to authentic resources facilitates the development of learners' language skills, enabling them to acquire proficiency in a more immersive and meaningful manner. Additionally, the use of technology has a positive impact on learners' motivation in language learning (Gikas and Grant 2013; Smith 2018). Peterson (2017) believes that technology empowers learners with a sense of autonomy and agency, allowing them to take control of their learning process.

With the advancements in artificial intelligence (AI) and natural language processing (NLP), intelligent tutoring systems and adaptive language learning platforms have emerged (Heift and Chapelle 2012). One such tool is ChatGPT, developed by OpenAI, which utilizes large-scale language models to generate human-like text based on a given input (Bender et al. 2021). ChatGPT demonstrates the ability in generating text, answering questions, and completing various language-related tasks (Kasneci et al. 2023). The utilization of ChatGPT holds significant potential for offering a wide range of benefits and opportunities for language learning across learners with different proficiency levels. According to Kasneci et al. (2023), ChatGPT can assist learners in developing language skills, such as writing and vocabulary acquisition, as well as providing personalized practice materials and explanations. Furthermore, ChatGPT can aid in tasks related to writing, research reports, and problem-solving while also offering discipline-specific language skills.

In addition to the positive effects of ChatGPT in language learning, there are also thought-provoking drawbacks related to its potential for cheating and its impact on assessment (Kohnke et al. 2023). One concern is that the responses generated by ChatGPT are paraphrases of its sources without proper citations, which can lead to issues such as plagiarism. Another drawback is that the text created by ChatGPT can be inconsistent. Kohnke et al. (2023) found responses generated by ChatGPT can vary significantly



when identical prompts are input multiple times, indicating the accuracy and reliability of its outputs depend heavily on how prompts are worded. Prompts are the initial text input to ChatGPT to provide context, while generated responses are the texts ChatGPT then produces.

The literature reviewed above largely consists of theoretical perspectives on the impact and effectiveness of tools such as ChatGPT in language learning. Additional research into learners' perspectives and understanding of these tools would further enrich our knowledge in this emerging field. While some initial studies have examined learner interaction with AI systems, there is still limited investigation regarding how students perceive ChatGPT specifically and how ChatGPT may facilitate them in completing language learning tasks.

### 2.2. The Potential of AI-Powered Chatbots

AI-powered chatbots, among various AI tools, have been recognized for their potential to deliver personalized and engaging language learning experiences. Chatbots, functioning as dialogue agents, engage in interactive conversations with human users, offering turn-by-turn interactions (Guo et al. 2022). These virtual conversation partners facilitate interactive language exchanges and provide learners with real-time feedback.

Various studies have explored the use of chatbots to support different aspects of language learning. By leveraging natural language processing (NLP) and machine learning techniques, chatbots can adapt to learners' proficiency levels and personalize the learning content. For example, Shin et al. (2021) developed an adaptive chatbot that adjusts the difficulty of conversations based on learners' language proficiency, resulting in improved engagement and learning outcomes. In the realm of pronunciation skills, Yang et al. (2022) found that learners who interacted with a pronunciation-focused chatbot demonstrated significant improvement in pronunciation accuracy compared to a control group. Chatbots also provide language learners with instant feedback, promoting learner autonomy and self-correction. Chen et al. (2022) used a chatbot to support learners' writing skills, offering automated feedback on grammar, vocabulary, and sentence structure, leading to improved writing performance over time. Additionally, Kim (2018) developed a vocabulary-focused chatbot that presented learners with targeted vocabulary items in context and provided immediate feedback on their usage, resulting in enhanced vocabulary retention. These studies collectively highlight the potential of chatbots to enhance language competence, provide personalized learning experiences, and foster positive learning outcomes in various language skills.

ChatGPT is an AI-powered chatbot capable of generating new content based on a large-scale language database. It differs from traditional question-and-answer chatbots by engaging in back-and-forth conversations with users. ChatGPT can summarize long articles or produce first drafts of presentations, providing new ideas for researchers and learners, and ChatGPT 4.0 even generates reasonably good quality academic articles on specific topics. It is important to note that ChatGPT is trained on existing resources, and its responses are based on a selection of those resources. Therefore, while it can rearrange and repeat existing information, users may need to adjust their prompts to obtain the desired responses. However, ChatGPT has limitations, including the potential to generate plausible but incorrect or made-up responses. This indicates that ChatGPT needs to be trained with high-quality data and the ability to understand complex prompts (Pavlik 2023). Additionally, language resources provided by ChatGPT are restricted to pre-2021 data, requiring users to critically evaluate responses derived from the chatbot.

In a nutshell, chatbots in language learning offer personalized experiences, adapt to learners' proficiency levels, and provide real-time feedback. ChatGPT, an AI-powered chatbot, generates new content but relies on existing resources. Users may need to adjust prompts to obtain desired responses, and critical evaluation is necessary due to potential inaccuracies and the chatbot's reliance on language resources predating 2021.

Building on the discussed importance of technology, there remains limited research exploring student perspectives and experiences with these emerging AI technologies, in particularly regarding the ChatGPT chatbot(Kasneci et al. 2023). To address this gap, the primary question of this study emerges:

R.Q.: How do learners perceive the role of ChatGPT in English language learning?

Investigating learners' perspectives is critical for several reasons. First, uncovering insights directly from students' first-hand experiences will provide key information on how ChatGPT addresses or fails to address learners' needs, preferences, and challenges. Gaining this user-centered understanding is essential to optimizing ChatGPT's utility for educational purposes. Second, learner perceptions influence acceptance and integration of the technology. Positive perceptions can facilitate the uptake of ChatGPT in language learning contexts, while negative perceptions may hinder its adoption. Finally, learner perspectives shape the ongoing ethical debates regarding AI; investigating student views contributes empirically grounded evidence to these discussions.

By exploring learner perceptions with ChatGPT, this study aims to gain insights that are actionable for developing effective applications of this technology that align with learners' values and goals. Understanding learners' perspectives is essential to optimizing the use of ChatGPT and addressing their specific needs and challenges in English language learning.

## 3. Research Methods

The current study employed a small-scale exploratory approach to investigate students' perceptions and experiences regarding the use of ChatGPT in the Chinese educational context.

### 3.1. Research Context

This study was conducted at a top-tier English-medium international university in China. The university did not officially advocate the integration of ChatGPT in teaching and learning at the time the data were collected; instead, it states specifically that by default ChatGPT should be prohibited in the completion of assignments that count towards the final grading of the course. However, students have gotten used to trying out different tools due to the open and inclusive academic atmosphere at the university. From the first author's observation, most students had experience using ChatGPT. Our study focused on students' experiences of and perceptions towards ChatGPT 3.5, which was the free version of ChatGPT which our participants used. Therefore, all ChatGPT mentioned thereafter refers to ChatGPT 3.5.

### 3.2. Research Participants

The participants were five undergraduate students from diverse majors at a Chinese university. Disciplines included Marketing, Translation, and Data Science across years 2–4, as shown in Table 1. The selection of participants aimed to capture diverse perspectives, considering that ChatGPT was still a relatively new concept in China. Participants were purposefully chosen based on the first author's classroom observations and regular interactions with students, ensuring all participants were active ChatGPT users. Specific selection criteria included: frequent ChatGPT usage strategies shared by Mike; Tessa's use of ChatGPT for IELTS preparation; Dylan's observed ChatGPT use for classwork; Derek's prior project on ChatGPT software; and Tere's own research on ChatGPT and feedback. This criteria-based, non-random sampling aimed to capture a diverse range of experiences with ChatGPT to allow for an information-rich, in-depth understanding of usage strategies and perceptions from the learners' perspective. Further details regarding the participants are provided in Table 1 below.

**Table 1.** Participants' information.

| Participants | Major | Year Groups |
|---|---|---|
| Mike | Marketing and communication | Year Four |
| Tessa | Translation | Year Three |
| Dylan | Data Science | Year Two |
| Derek | Data Science | Year Two |
| Tere | Translation | Year Three |

### 3.3. Data Collection and Analysis

Data for this study were collected through semi-structured interviews focusing on students' views and experiences regarding the use of ChatGPT for language learning The interview protocol can be found in the Appendix A. Guided by the theoretical review conducted by Kohnke et al. (2023), the interviews primarily addressed three key issues: (1) students' knowledge and understanding of ChatGPT; (2) how students utilized ChatGPT in language learning; and (3) students' awareness of handling the drawbacks and challenges associated with the use of ChatGPT. The interviews were conducted in Chinese, the participants' mother tongue, and were transcribed by trained research assistants. All interviews were audio recorded with the participants' consent. Ethical approval for this study was obtained from the university affiliated with the first author.

Data analysis were conducted thematically using NVivo 11, guided by the thematic map of the power of ChatGPT developed by Yan (2023). This thematic map encompassed three key aspects: (1) the power of ChatGPT; (2) potential challenges; and (3) the proper use of ChatGPT. Additionally, a fourth theme, "critical reflections on ChatGPT", emerged from the findings. Yan's (2023) thematic maps guide the direction of our data analysis, with detailed codes emerging from the data itself. For example, codes coded under the power of ChatGPT included "generating ideas", "providing individualized assistance", and "offering immediate feedback". The data were initially coded by the first author and subsequently reviewed and confirmed by the second author.

## 4. Findings

### 4.1. ChatGPT as a Peer Tutor for Providing Individualized Assistance

Three out of five students described ChatGPT as a peer tutor that provided support which was more easily accessible than that of teachers. Tessa, for instance, used ChatGPT as a learning partner while preparing for the writing test in IELTS, as she noted:

> I asked ChatGPT how I could improve the logic of my essay, and it will tell me which sentences are not well connected and the information I need to connect ideas. I also asked ChatGPT to add additional examples to make sure the texts are well connected. (Tessa)

This excerpt indicates that ChatGPT played the role of a language tutor by engaging in communication with students to facilitate improvement. Tessa mentioned that she now visited the Language Centre less frequently since ChatGPT offered instant feedback on multiple pieces of writing; additionally, ChatGPT can even offer feedback regarding how to improve the essay so that it can achieve a score of Band 7 or 8. It seems that the ability to offer instant and easily accessible feedback is a key attractive feature of ChatGPT.

Similarly, Dylan shared that ChatGPT tutored him on developing different sections of a paper, similar to teachers. He mentioned that ChatGPT provided instructions on writing an effective "Introduction," "Problems," and "Solutions" section in a technical proposal; therefore, he did not need to ask teachers for clarifications. Derek remarked that the ChatGPT tutor has the ability to offer personalized feedback on whatever confused him. He appreciated how he could ask step-by-step questions to learn how to develop an essay in authentic English with convincing examples. Such responses support the idea that ChatGPT offers personalized and adaptive learning experiences (Heift and Chapelle 2012; Guo et al. 2022).

Regarding whether the assistance of ChatGPT helps with developing real language competence, both Tessa and Dylan affirmed that their interactions with ChatGPT had helped them improve their grammar, vocabulary, and essay coherence. This was partially because both of them were preparing for an international language test that they needed to take by themselves. Thus, they used ChatGPT more as a resource for assistance. However, all students admitted that ChatGPT contributed less or even restricted the development of language competence if students merely used it as a ghostwriter to complete the assignments for them. The extent to which ChatGPT could possibly contribute to language development was unsurprisingly tied to whether it was used as an assistant or a replacement. Tessa further noted that ChatGPT primarily assisted with lower-order language skills, such as basic vocabulary and grammar, while providing less support for advanced language skills, such as writing logical and authentic essays. Derek expressed the belief that language skills are better developed through extensive reading and listening rather than solely relying on ChatGPT interactions. According to him, ChatGPT can improve the appearance of language but falls short of developing true language competence. These student responses highlighted that ChatGPT has the potential to enhance certain language skills only when used appropriately.

### 4.2. ChatGPT as a Source for Generating New Ideas

All five participants unanimously agreed that they would utilize ChatGPT to generate new ideas when planning or writing an English essay, as shown below:

> ChatGPT helps with brainstorming ideas. For example, if I plan to write an essay on a specific topic. I will ask ChatGPT to give me some ideas about where to start. (Dylan)

> When I do not know where to start, I will ask ChatGPT to think of several topics for me; then, I will take a look to see which one I am interested in. (Mike)

These responses demonstrated how ChatGPT contributed to the planning stage when students completed a language writing task. Tere added that although the ideas generated by ChatGPT were not necessarily innovative, they reminded her of ideas she had not come up with on her own. Tere further elaborated that the ideas generated by ChatGPT were somewhat broad and vague; therefore, she used them as a starting point and then looked for examples herself. Tere's answer shows her awareness of both the strengths and drawbacks of ChatGPT in generating new ideas and her attempt to make use of its strengths.

Derek believed that ChatGPT helped a lot with generating ideas; however, he felt that this was not something new, as he used Google to search for new ideas before the birth of ChatGPT:

> Generating new ideas through ChatGPT is not something new. Without ChatGPT, I obtained new ideas through other means, such as reading books or searching Google. ChatGPT just made the whole process more efficient by integrating all sorts of information. (Derek)

Derek's response seemed to indicate that he only regarded ChatGPT as a technology tool that was more efficient than other tools he had used before.

### 4.3. Revising Prompts to Maximize the Learning Effects

All participants reached a consensus that they needed to constantly modify the prompts to obtain useful and accurate information. Tere described ChatGPT as a child whom she needs to teach:

> I cannot simply say: please write an English essay on something. I need to provide detailed instructions and explain the sorts of academic styles needed, ideas I already have, my expectations, or even the specific topics. ChatGPT can then generate something you need. (Tere)

Tere was clearly aware that ChatGPT needed constant training to function effectively, and she was aware of the successful strategies for a quality essay. Tessa explained that she sent her essay paragraph by paragraph to ChatGPT, seeking specific assistance such as adding examples and enhancing criticality and logic, as ChatGPT worked out better when it was provided with specific instructions on working on a short paragraph. Derek, who was working on a ChatGPT-related project with his data science professor, noted that data science professors were working on generating optimized prompts that could better meet the needs of users. For example, instead of asking ChatGPT not to perform something, it is better to state in a positive way what ChatGPT should do. The responses from the three students highlight the significance of using effective prompts to achieve productive outcomes.

Regarding how students obtained the knowledge of effective prompts, they responded that they did this through constant trials, social media, or exchanging ideas with peers, as Tessa noted:

> There are a lot of posts on Xiao Hongshu (a social media app) about guiding Chat-GPT to generate effective prompts. These posts provide examples of prompts and keywords that users can use to ask ChatGPT to generate useful content. (Tessa)

Such a response seemed to indicate that students viewed ChatGPT as a technology-enhanced tool and were actively engaged in exploring strategies and techniques to maximize the benefits of ChatGPT in helping them produce a better essay in English.

### 4.4. Making Critical Judgments on Information Generated by ChatGPT

Whereas ChatGPT appeared to be effective in generating ideas and providing assistance, students reflected critically on the information generated by ChatGPT instead of accepting it automatically, as Mike explained in the quotes below:

> I always took a careful look at what ChatGPT produced. I once asked ChatGPT to draft an essay on a novel, but the outcome reads so unrealistic that I doubted if it had ever read this novel. Then I gave it up and tried to come up with an essay by myself. (Mike)

The response above indicated Mike was aware that ChatGPT does not necessarily generate accurate information. Similarly, Dylan and Tere reported that ChatGPT seldom said no to any questions and always made things up; therefore, they needed to judge if the information generated was usable.

To address the issue of inaccurate information, Tessa opted to verify the information by consulting dictionaries or more reliable sources. As an example, she mentioned that she once asked ChatGPT for synonyms for "be closely related to." After receiving the response, she cross-referenced dictionaries and corpora to confirm the word's appropriateness in her specific context. Such a response shows that students only used ChatGPT as a source of information, and they still held critical attitudes toward its actual effectiveness.

Apart from consulting authoritative resources, Dylan mentioned that he would engage in multiple conversations by asking the same question repeatedly. He would then compare the responses from these interactions to assess the accuracy of the information generated.

Even when ChatGPT generated useful or accurate information, students did not automatically accept all of it, as illustrated by Dylan:

> When I was working on my technical proposal on improving the face recognition system, ChatGPT generated much useful advice, such as improving the database, algorithm, and hardware, etc. I did not accept all the ideas, which were too complex. I chose to focus on the hardware only and worked a bit further on the hardware. (Dylan)

Dylan's response further affirms that ChatGPT was only used as a learning tool rather than a substitute for learning.

However, instead of consistently checking the accuracy of the information, Mike reported that he chose not to rely on ChatGPT for important tasks as it was really hard to instruct ChatGPT to perform the task as he expected:

> I do not really believe that ChatGPT can really understand what I was talking about. For example, when I asked it to re-write my sentence in a native or authentic way. I doubt if it understood what native or authentic means. In doing translation, I also do not believe ChatGPT can act like a human translator by following the principle of "faithfulness", "expressiveness", and "elegance" in a translation task. (Mike)

Among the five participants, Mike held a comparatively negative view towards ChatGPT, as he believed that ChatGPT was still a program instead of a real human who was capable of understanding his intended purposes. He explained that in his experience preparing documents for postgraduate applications, he felt ChatGPT did not really know what he needed, and the writing could not truly reflect his personality.

All in all, all participants seemed to hold a critical view of the effectiveness of ChatGPT. Students' abilities to critically reflect on the capacity of ChatGPT seemed to be related to three resources: their understanding of the nature of ChatGPT, their own interactions and experimentation with ChatGPT, and the guidance provided by their teachers. Three science students, Mike, Dylan, and Derek, kept emphasizing that ChatGPT was essentially a language model based on Natural Language Processing, lacking the ability to truly understand or comprehend user input. Additionally, teachers' guidance seemed to be equally important. For example, Tere mentioned that her translation teacher encouraged them to compare AI-generated translations with human translations, leading them to realize that ChatGPT could only provide direct translations without considering contextual nuances. These findings suggest that students are more likely to use ChatGPT effectively when they have a better understanding of its limitations and capabilities.

*4.5. ChatGPT and Plagiarism Issues*

Regarding the potential plagiarism issues caused by ChatGPT, all participants seemed to be aware of such a potential threat, as evident in Dylan's response below:

> Students would always think since you have already brainstormed the ideas for me, why not write an essay for me. Yes, this is different from the intended purpose of education, but those students who just wanted to complete the homework will do this. (Dylan)

Derek shared a similar view, noting that whether students used the tool appropriately depended on whether they considered the completion of the task important. For students with low motivation to learn, it is unavoidable that they would take a shortcut by using ChatGPT as a ghostwriter.

However, all students felt that it was impossible to ban students from using ChatGPT. They believed that it was the course instructors' responsibility to rethink what skills they should teach in the new technology era and guide students to use such a tool in a legitimate way.

> I think being able to use ChatGPT is also kind of ability, just like the ability to Google. No one will doubt us when we obtain ideas of others from Google searches. I believe ChatGPT will become a tool like Google in the future. In the future, when we write our Resume, we can simply use ChatGPT; then why do we need to learn the skills of writing a resume ourselves? (Derek)

Derek's response seemed to bring out a challenging issue about what to teach in the new AI era. Many students used a similar example: since AI programmers can work much faster and more efficiently compared with human programmers, it will be a trend that many coding tasks in the future can be completed by AI.

Additionally, students believed that proper guidance from teachers was equally important:

I think it is hard to forbid students to use ChatGPT. I think teachers should guide students to use it in a positive way. (Tessa)

I think we should tell students that they need to report how they use ChatGPT. Students can be encouraged to use ChatGPT as long as they clearly reported how ChatGPT was used. It is a good reflective process. (Tere)

The responses above further supported the idea that one potential strategy to handle the threat of plagiarism is to guide students to use ChatGPT in a legitimate and productive way. Students' views reaffirm that future teachers need to consider ChatGPT as a tool instead of a threat, integrating it into their pedagogical practices to promote effective and responsible use.

## 5. Discussion and Conclusions

The current study represents one of the first few empirical investigations exploring students' experiences with ChatGPT shortly after its introduction and in what way ChatGPT supports/restricts language learning. In response to our research question, students identified three key benefits associated with ChatGPT usage. Firstly, ChatGPT functions as a learning partner or personal tutor by providing personalized, easily accessible, and adaptive feedback. Secondly, it assists in improving language proficiency when students apply critical thinking skills, such as modifying prompts, training the model, and verifying and selectively accepting its outputs. Thirdly, it facilitates idea generation for brainstorming purposes. Most importantly, our students demonstrated the capacity to think critically about the information generated by ChatGPT and reported their ability to modify prompts, train ChatGPT, verify, and selectively accept the information provided. A significant argument arising from this study is the necessity for teachers and educators to embrace this new AI tool and offer appropriate guidance to students.

Our study addresses a significant topic concerning the utilization of technology-enhanced AI tools in language teaching and learning. While debates regarding the impact of ChatGPT on language learning exist (Kohnke et al. 2023; Yan 2023), our findings contribute positively by supporting the notion that ChatGPT can serve as an effective tool for providing immediate feedback and personalized learning experiences (Chen et al. 2022; Kim 2018). In this regard, ChatGPT functions as a personal language tutor for students. The participants in our study sought assistance from ChatGPT in tasks such as text revision, structural and content suggestions for essays, and the provision of examples to support their arguments. These findings suggest that ChatGPT holds the potential to act as a tutor, particularly in large classrooms where teachers are not that easily accessible.

While our study did not directly collect evidence from students' writings, participants reported that the assistance of AI tools, such as ChatGPT, contributed to the improvement of textual quality (Yan 2023). Moreover, our findings suggest that proper utilization of ChatGPT has the potential to enhance language competence. Tessa and Dylan, for instance, mentioned their ongoing reflection on essay improvement as they prepared for external international assessments. This reflective process not only aids in enhancing language skills but also promotes student autonomy (Kasneci et al. 2023; Kohnke et al. 2023), as students actively engage in self-revision based on the suggestions provided by ChatGPT.

Whereas ChatGPT generated a lot of useful information, a notable finding in our study is that students developed a critical stance towards the role of ChatGPT in English language learning, which has not yet been explicitly addressed in existing literature. Participants did not blindly accept the information generated by ChatGPT but instead evaluated its accuracy, relevance, and specificity. Whereas previous studies identified opportunities for designing tasks that develop students' critical thinking skills through ChatGPT (Kasneci et al. 2023), students in our study seemed to have developed the ability to think critically about the strengths and drawbacks of ChatGPT without explicit training. The data suggests that this critical thinking ability is probably related to their rich experiences with ChatGPT and the trial-and-error process they underwent. Students consistently critiqued the limited potential of AI in various aspects of learning and emphasized the need to

judge if the information is usable. Such findings form a sharp contrast with Kasneci et al.'s (2023) worries about students' over-reliance on ChatGPT and its potential to replace traditional learning. To address the potential threats posed by ChatGPT and promote its productive use, teachers should ensure students are aware of the limitations of language models and encourage them to take responsibility for their own learning (Pavlik 2023).

Interestingly, whereas students in Yan's (2023) study revealed more worries and concerns over the versatility of ChatGPT, for example, its danger to academic integrity, students in our study took a more positive attitude, and all but one student strongly believed that universities should embrace, rather than ban, the use of ChatGPT. Such an open attitude might be related to the rich experience students have had with ChatGPT as well as their capacity to view ChatGPT critically. It appears that their concerns regarding the potential threats of ChatGPT were mitigated by their critical judgment of its potential benefits.

Our study generated pedagogical implications for the use of ChatGPT in English language teaching. First, our findings support the possibility of integrating technology-enhanced tools, such as ChatGPT, into the language classroom. It must be admitted that it is an unavoidable trend that we are moving towards a new AI era, and the pedagogical tools need to be revolutionized accordingly. Second, our findings highlight the importance of providing pedagogical guidance to help students utilize ChatGPT in a legitimate, proper, and productive manner (Yan 2023), instead of banning the tool completely.

The methodological limitation of this study lies in the fact that it only collected data from a small number of students shortly after the birth of ChatGPT It is possible that students' perceptions might be different if they received systematic training or guidance instead of exploring things by themselves. This exploratory study only provides initial insights that require further research among larger and more diverse samples. Future studies could address this limitation by incorporating a well-designed training program to investigate students' experiences and perceptions in a more comprehensive manner.

**Author Contributions:** Conceptualization, Y.Z. and Y.X.; methodology, Y.X.; software, Y.X.; validation, Y.Z.; formal analysis, Y.X.; investigation, Y.X. and Y.Z.; resources, Y.X.; data curation, Y.X.; writing—original draft preparation—Y.X. and Y.Z.; writing—review and editing, Y.Z. and Y.X.; project administration, Y.X. and Y.Z.; funding acquisition, Y.X. and Y.Z. All authors have read and agreed to the published version of the manuscript.

**Funding:** The research was funded by the University of Shanghai for Science and Technology, grant number CFTD2023YB22; Shanghai International Studies University, grant number Z-2023-305-013; and Shenzhen Educational Sciences Planning Scheme (the 14th Five-Year Plan), grant number YBZZ21019; Teaching Innovation Grant, The Chinese University of Hong Kong, Shenzhen, grant number I10120230444.

**Institutional Review Board Statement:** The Ethical Approval of this Study has been obtained from The Chinese University of Hong Kong, Shenzhen, Approval code: CUHKSZ-D-20230015.

**Informed Consent Statement:** Informed consent was obtained from all subjects involved in this study.

**Data Availability Statement:** The data supporting the findings of this study are available upon reasonable request.

**Conflicts of Interest:** The authors declare no conflict of interest.

## Appendix A. Interview Protocol

### Using ChatGPT in language learning: Students' perspectives.

a.  How much do you know about ChatGPT? What do you usually use Chat GPT for?

你对ChatGPT的了解如何？你用了多久的ChatGPT呢？平时你主要会用ChatGPT来干什么呢？

b.  What are your teachers' attitudes towards ChatGPT? For example, do you have any experience where teachers encouraged the use of ChatGPT in a specific class? Can you provide me with examples?

你的老师们对ChatGPT的态度是什么样的呢？你是否有一些老师会鼓励大家使用ChatGPT, 你们是如何使用的呢？

c.  Could you please briefly describe your experience with ChatGPT in language learning? Can you give me one specific example?

可以介绍下在语言学习方面，你通常会用ChatGPT来做什么呢？可以介绍一个和语言学习相关的具体例子吗？

d.  Compared with how you studied English before, what do you think are the particular strengths of ChatGPT in language learning? Can ChatGPT help/facilitate you to study independently?

相比于你过往的语言学习方法，你认为ChatGPT在语言学习方面有什么特别的优势吗？ChatGPT可以帮助你更好的自主学习吗？

e.  What specific language skills do you think ChatGPT may provide more support with?

如果考虑到不同的语言技能（如：听说读写词汇语法），你认为ChatGPT在哪个方面会更有用呢？

f.  Could you provide examples of effective prompts you have used when interacting with ChatGPT?

当你和ChatGPT互动的时候，你通常会给ChatGPT提供一些什么样的prompt呢？
当ChatGPT不能回答你的提问的时候，你是如何修改你的prompt的呢？

g.  How do you judge if the resources provided by ChatGPT are accurate or reliable?

你是如何判断ChatGPT提供的resource是否准确或者可信呢？

h.  What challenges have you met when you used ChatGPT? How did you troubleshoot challenges when using ChatGPT in the classroom?

你在使用ChatGPT的时候遇到过什么问题吗？你是如何解决的呢？

i.  What can be the major drawbacks of the use of ChatGPT?

ChatGPT的使用有什么缺点吗？

j.  How do you think about the potential challenges related to academic integrity and ethical issues?

你如何看待ChatGPT可能导致的学术诚信和伦理问题呢？

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
