# Peer review of "An Exploratory Study of EFL Learners’ Use of ChatGPT for Language Learning Tasks: Experience and Perceptions"

_languages, doi:10.3390/languages8030212_

Round 1
Reviewer 1 Report
General comment:
This is an interesting and very relevant study as there is an urgent need to understand how to best integrate ChatGPT in language learning and teaching activities. However, on top of the relatively minor comments I list below, the authors need to more clearly frame the study. Based on the title and introduction, it sounds like the study focuses on language learning. However, in the results and discussion sections, it is unclear how the different aspects analysed in sections 4.2-4.5 relate to language learning specifically (rather than to learning and essay writing at university in general). Moreover, more information is needed about the participants in terms of their linguistic background, where ChatGPT intervenes in their language learning, etc…
By section:
Abstract:
-could the authors clarify what they mean by “appropriateness”?
-could the authors add the number of participants?
-l.12-15: are these ideas the participants said? If so, this need to be clarified?
-“participants exhibited critical judgment in evaluating ideas and outputs generated by ChatGPT ”ïƒ could the authors give an example of what these critical judgments were? Otherwise, this statement sounds a bit vague
Introduction:
-“Another drawback is the accuracy of the generated texts since the generated 90 responses may vary if the prompts are regenerated (Kohnke et al., 2023).”ïƒ This sentence might be difficult to understand for someone not familiar with ChatGPT. Could it be reformulated? What are “regenerated responses”? What are prompts?
-from the paragraph l.92-97, it is unclear whether there already exists some research regarding the “understanding of these tools from the learners’ perspective”ïƒ could this be clarified?
Methods:
-In the research participants section, it is not always clear how the way the students used ChatGPT related to their language learning. It would also be useful to give an overview of the participants’ proficiency level in English.
-Could the authors provide the actual interview questions?
Findings:
-it is not clear how sections 4.2 and 4.3 relate to language learning itself.
Discussion:
- “Secondly, it assists in improving language proficiency when used appropriately.”: what does “appropriately” mean?
Line-by-line:
-l.25: “to support language learning has learning has attracted”: there seems to be a typo here
-l.223: “affirmed affirmed”; l.225: “both both”
The quality of the language is fine; I only spotted minor typos.
Author Response
Reviewer 1
General comment:
This is an interesting and very relevant study as there is an urgent need to understand how to best integrate ChatGPT in language learning and teaching activities. However, on top of the relatively minor comments I list below, the authors need to more clearly frame the study. Based on the title and introduction, it sounds like the study focuses on language learning. However, in the results and discussion sections, it is unclear how the different aspects analysed in sections 4.2-4.5 relate to language learning specifically (rather than to learning and essay writing at university in general). Moreover, more information is needed about the participants in terms of their linguistic background, where ChatGPT intervenes in their language learning, etc…
By section:
Abstract
Comment 1: -could the authors clarify what they mean by “appropriateness”?
Response 1: We have modified this sentence to “However, to what extent ChatGPT can be used for assisting language learning remains largely unexplored.”, as we realized that appropriateness is not a main focus of the research.
Comment 2: -could the authors add the number of participants?
Response 2: Data were collected through semi-structured interviews with five students in a top-tier international university in China.
Comment 3 -l.12-15: are these ideas the participants said? If so, this need to be clarified?
Response 4: Thanks a lot. We have added a few words to indicate that these are the participants’ responses, as shown below:
Students’ responses revealed that ChatGPT has the potential to serve as a valuable learning partner and aid students in developing language abilities. Furthermore, participants exhibited critical judgment in evaluating the quality and reliability of ideas and outputs generated by ChatGPT, as well as the ability to modify prompts to maximize learning benefits.
Comment 4 -“participants exhibited critical judgment in evaluating ideas and outputs generated by ChatGPT ”àcould the authors give an example of what these critical judgments were? Otherwise, this statement sounds a bit vague
Response 4 – We tried to modify this part to: “Furthermore, participants exhibited critical judgment in evaluating the quality and reliability of ideas and outputs generated by ChatGPT”. As this is only the abstract, more details regarding the critical judgement will be elaborated in the finding and discussion section.
Introduction:
Comment 5 -“Another drawback is the accuracy of the generated texts since the generated 90 responses may vary if the prompts are regenerated (Kohnke et al., 2023).”àThis sentence might be difficult to understand for someone not familiar with ChatGPT. Could it be reformulated? What are “regenerated responses”? What are prompts?
Response 5 Here is one way to rephrase the sentence to make it more understandable:
“Another drawback is that the text created by ChatGPT can be inconsistent. Kohnke et al. (2023) found ChatGPT's generated responses can vary significantly when identical prompts are input multiple times, indicating the accuracy and reliability of its outputs depend heavily on how prompts are worded. Prompts are the initial text input to ChatGPT to provide context, while generated responses are the texts ChatGPT then produces.”
Comment 6 -from the paragraph l.92-97, it is unclear whether there already exists some research regarding the “understanding of these tools from the learners’ perspective” could this be clarified?
Response 6 Thanks for this comment and we rewrite this paragraph to soften the claim:
"The literature reviewed above largely consists of theoretical perspectives on the impact and effectiveness of tools like ChatGPT in language learning. Additional research into learners’ perspectives and understanding of these tools would further enrich our knowledge in this emerging field. While some initial studies have examined learner interaction with AI systems, there is still limited investigation regarding how students perceive ChatGPT specifically and integrate it into their learning processes. To effectively utilize ChatGPT in language education, exploring how learners view this technology and addressing their needs and challenges through empirical evidence will be crucial steps in complementing the current theoretical foundations."
Methods:
Comment 7
-In the research participants section, it is not always clear how the way the students used ChatGPT related to their language learning. It would also be useful to give an overview of the participants’ proficiency level in English.
Response 7
We have modified the title of our paper to: An exploratory study of EFL learners’ use of ChatGPT for language learning tasks: Experience and Perceptions. As the key focus of our paper is to explore how students used ChatGPT to complete language-related tasks, such as essay writing, we did not study how ChatGPT affected students’ language proficiency level. Therefore, information regarding their language proficiency level was not collected. Most participants can get around 7-8.5 in IELTS. In general, their language proficiency is high.
Comment 8
-Could the authors provide the actual interview questions?
Response 8
Yes, of course. We have included the interview questions in Appendix.
Findings:
Comment 9
-it is not clear how sections 4.2 and 4.3 relate to language learning itself.
Response 9:
We agreed that these two sections were not explicitly related. We tried to add a few words to indicate that students used ChatGPT to complete language-related tasks. For example
- All five participants unanimously agreed that they would utilize ChatGPT 3.5 to generate new ideas when planning or writing an English essay, as shown below:
- These responses demonstrated how ChatGPT 3.5 contributed to the planning stage when students completed a writing task.
- I cannot simply say: please write an English essay on something. I need to provide detailed instructions and explain the sorts of academic styles needed, ideas I already have, my expectation, or even the specific topics.
- Tere was clearly aware that ChatGPT 3.5 needed constant training to function effectively and she was aware of the successful strategies for a quality essay. Tessa explained that she sent her essay paragraph by paragraph to ChatGPT 3.5, seeking specific assistance like adding examples and enhancing criticality and logic.,
Discussion:
Comment 10
“Secondly, it assists in improving language proficiency when used appropriately.”: what does “appropriately” mean?
Response 10
Thanks for your question. Yes, the word “appropriately” is too abstract to understand. We rewrite the statement as the following:
“Secondly, it assists in improving language proficiency when students apply critical thinking skills, such as modifying prompts, training the model, and verifying and selectively accepting its outputs.”
Line-by-line:
Comment 11
-l.25: “to support language learning has learning has attracted”: there seems to be a typo here
-l.223: “affirmed affirmed”; l.225: “both both”
Response 11
Yes,they are typos and many thanks for pointing out and we have corrected them in the revised version.
Reviewer 2 Report
Comments and suggestions for authors
Overall
ChatGPT and other LLMs are the latest disruptive technology to impact language learning and so research exploring learner experiences of using LLMs is particularly pertinent. The authors conducted a small-scale case study on university student usage of ChatGPT for language learning purposes. The authors claim that the benefits to learners outweigh the drawbacks to academic integrity. The authors rightly note that there is little research from a practical perspective on the use of LLMs, which is unsurprising given that access to ChatGPT only started to spread widely at the beginning of 2023. The authors report the students’ views interspersed with verbatim quotes and authorial insights. This is likely to be the first of an avalanche of such papers.
Major issues
1. Research question
This research question suddenly appears. Why did the researchers decide to focus on learner perceptions? There are many possible areas of investigation, such as learner achievement, learner motivation and learner engagement. I am not suggesting the authors should change their focus, but simply show how the decision to focus on perceptions was made, and why this is of importance.
2. Sample selection
Case studies, given their small-scale, are intrinsically biased. However, the authors should still declare how the sample of 5 students was selected. They did not appear to be randomly selected.
3. Thematic analysis
There is a paucity of information on the thematic analysis phase. Bar a passing mention to NVivo 11 and the work on Yan (2023). Does this mean that the authors did not conduct their own thematic analysis but simply followed Yan and added one more theme. The authors should provide more clarity on this phase.
Minor issues
Line 223 affirmed affirmed
Lines 333-334 ChatGPT is ChatGPT is
Line 337 compare compare
Various spacing issues, e.g. Line 395 Kim,2018
Author Response
Reviewer 2
Comments and suggestions for authors
Overall
ChatGPT and other LLMs are the latest disruptive technology to impact language learning and so research exploring learner experiences of using LLMs is particularly pertinent. The authors conducted a small-scale case study on university student usage of ChatGPT for language learning purposes. The authors claim that the benefits to learners outweigh the drawbacks to academic integrity. The authors rightly note that there is little research from a practical perspective on the use of LLMs, which is unsurprising given that access to ChatGPT only started to spread widely at the beginning of 2023. The authors report the students’ views interspersed with verbatim quotes and authorial insights. This is likely to be the first of an avalanche of such papers.
Major issues
Comment 1. Research question
This research question suddenly appears. Why did the researchers decide to focus on learner perceptions? There are many possible areas of investigation, such as learner achievement, learner motivation and learner engagement. I am not suggesting the authors should change their focus, but simply show how the decision to focus on perceptions was made, and why this is of importance.
Response 1
We appreciate the reviewer raising this important point. Our decision to focus specifically on learner perceptions of ChatGPT was made for several key reasons:
First, as noted in our literature review, there is limited prior research exploring student perspectives and experiences with these emerging AI technologies from the learner standpoint (Kasneci et al., 2023). Investigating learner perceptions helps address this gap by uncovering insights directly from students' first-hand experiences using ChatGPT for language learning.
Second, we argue that learner perspectives are critical to understand, in order to optimize ChatGPT's utility for educational purposes. Uncovering learner perceptions and pain points provides key insights into how ChatGPT addresses or fails to address students' needs, preferences, and challenges. This understanding of the subjective user experience will be essential for developing effective applications of ChatGPT that align with learners' values and goals.
Finally, learner perceptions are important because negative perceptions may hinder acceptance and adoption of the technology, while positive perceptions can facilitate its integration. As ChatGPT diffuses into educational contexts, it will be critical to understand student perspectives to foresee and address any tensions that may arise.
In sum, we believe our focus on learner perceptions is justified given the gap in the literature, the value of a user-centered perspective, and the influence of perceptions on ChatGPT's educational integration. However, we agree with the reviewer that achievement, motivation, engagement and other areas are also rich directions for further research on ChatGPT's applications in language learning. Thus, we rewrite this section as below:
“Building on the discussed importance of technology, including ChatGPT chatbots, in language learning, there remains limited research exploring student perspectives and experiences with these emerging AI technologies from the learner standpoint (Kasneci et al., 2023). To address this gap, the primary question of this study emerges:
R.Q.: How do learners perceive the role of ChatGPT in English language learning?
Investigating learner perspectives is critical for several reasons. First, uncovering insights directly from students' first-hand experiences will provide key information on how ChatGPT addresses or fails to address learners' needs, preferences, and challenges. Gaining this user-centered understanding is essential to optimize ChatGPT's utility for educational purposes. Second, learner perceptions influence acceptance and integration of the technology. Positive perceptions can facilitate uptake of ChatGPT in language learning contexts, while negative perceptions may hinder its adoption. Finally, learner perspectives shape the ongoing ethical debates regarding AI; investigating student views contributes empirically grounded evidence to these discussions.
By exploring learner perceptions with ChatGPT, this study aims to gain insights that are actionable for developing effective applications of this technology that align with learners' values and goals. Understanding learners' perspectives is essential to optimize the use of ChatGPT and address their specific needs and challenges in English language learning.”
Comment 2. Sample selection
Case studies, given their small scale, are intrinsically biased. However, the authors should still declare how the sample of 5 students was selected. They did not appear to be randomly selected.
Response 2
We appreciate the reviewer raising this important point about sample selection. Given the intrinsic biases in small-scale qualitative studies, we agree it is imperative to provide full details on how our sample of 5 students was chosen. In the revised manuscript, we have added more context to the Research Participants section on the purposeful selection methods used:
"Participants were purposefully chosen based on the first author's classroom observations and regular interactions with students, ensuring all participants were active ChatGPT users. Specific selection criteria included: frequent ChatGPT usage strategies shared by Mike; Tessa's use of ChatGPT for IELTS prep; Dylan's observed ChatGPT use for classwork; Derek's prior project on ChatGPT software; and Tere's own research on ChatGPT for feedback. This criteria-based, non-random sampling aimed to capture a diverse range of experiences with ChatGPT to allow for an information-rich, in-depth understanding of usage strategies and perceptions from the learner perspective."
Comment 3. Thematic analysis
There is a paucity of information on the thematic analysis phase. Bar a passing mention to NVivo 11 and the work on Yan (2023). Does this mean that the authors did not conduct their own thematic analysis but simply followed Yan and added one more theme? The authors should provide more clarity on this phase.
Response 3.
Yan’s (2023) thematic maps guide the direction of our data analysis, with detailed codes emerging from the data itself. For example, codes coded under the power of ChatGPT included: “generating ideas”, “providing individualized assistance”, and “offering immediate feedback”.
Comment 4:Minor issues
Line 223 affirmed affirmed
Lines 333-334 ChatGPT is ChatGPT is
Line 337 compare compare
Various spacing issues, e.g. Line 395 Kim,2018
Response 4
Thanks for pointing out these typos and they were corrected in the revised manuscript.
Reviewer 3 Report
Detailed comments were attached.

Author Response
I reviewed the study “ChatGPT in language learning: An exploratory study of EFL learners’ experience of using ChatGPT”. My final decision is to reject the study according to the following comments.
In terms of findings and empirical evidence, nothing was seen to support the scientific contribution in the study. For example;
Comment 1: “Data were collected through semi-structured interviews in a top-tier international university in China” There is nothing about the collected data. Which kinds of data were collected? What are the data attributes? What were the questions and answers? (A sample could be given)
Response 1: We have mentioned in our research method section that data were collected through semi-structured interviews. Data were students’ interview responses.
- Based on your suggestion, we have included our interview questions in Appendix One.
- We also elaborated on how we coded the data, as shown below:
Yan’s (2023) thematic maps guide the direction of our data analysis, with detailed codes emerging from the data itself. For example, codes coded under the power of ChatGPT included: “generating ideas”, “providing individualized assistance”, and “offering immediate feedback”.
- Students’ responses can be found in the direct quotes in the finding section.
Comment 2: “Our findings contribute to the understanding of the potentials of ChatGPT in language education by adding empirical evidences from students’ perspectives” What is the empirical evidence? There is no presentation or explanation about the empirical evidence.
Response 2: In qualitative study, students’ quotations can be considered as empirical evidence. Our findings and key themes emerged from the thematic analysis of students’ responses, which can be represented in the quotes.
Comment 3: “Such findings generate implications on the future pedagogical practices in the new era …” Findings obtained from only five students are not enough to generate future pedagogical practices. I suggest using at least statistical methods to support the findings.
“Participants of this study were five undergraduate students from different majors” The authors interviewed with only five students and tried to generalize the study’s conclusion with this too limited research.
Comment 3: We understand that statistical research can generate more representative findings. As our study is an exploratory study, we felt that a qualitative study with a small sample of five students can satisfy the needs of the study, in particular considering that this study was conducted when ChatGPT was just released. An exploratory study will help us unpack students’ perceptions, which can be used for large-scale quantitative studies later.
Comment 4: The scientific contribution of the study should be clearly given in the abstract and introduction.
Comment 4: We have clarified our contribution to the introduction section.
Our findings contribute to the understanding of the potentials of ChatGPT in language education by adding empirical evidences from students’ perspectives. The study supports that ChatGPT can work as an effective tool for providing students with immediate feedback and personalized learning experiences.
Comment 5: Motivation should be mentioned. What was the problem? What was the missing part of the solutions proposed by previous studies?
Response 5: We have clarified and emphasized the gap as shown below:
Building on the discussed importance of technology, including ChatGPT chatbots, in language learning, there remains limited research exploring student perspectives and experiences with these emerging AI technologies from the learner standpoint (Kasneci et al., 2023). To address this gap, the primary question of this study emerges:
R.Q.: How do learners perceive the role of ChatGPT in English language learning?
Investigating learner perspectives is critical for several reasons. First, uncovering insights directly from students' first-hand experiences will provide key information on how ChatGPT addresses or fails to address learners' needs, preferences, and challenges. Gaining this user-centered understanding is essential to optimize ChatGPT's utility for educational purposes. Second, learner perceptions influence acceptance and integration of the technology. Positive perceptions can facilitate uptake of ChatGPT in language learning contexts, while negative perceptions may hinder its adoption. Finally, learner perspectives shape the ongoing ethical debates regarding AI; investigating student views contributes empirically grounded evidence to these discussions.
By exploring learner perceptions with ChatGPT, this study aims to gain insights that are actionable for developing effective applications of this technology that align with learners' values and goals. Understanding learners' perspectives is essential to optimize the use of ChatGPT and address their specific needs and challenges in English language learning.
Reviewer 4 Report
The general idea of the research sounds interesting. It is an exploratory study based on a small number of participants (5 people) so the outcomes may not be relevant to a broader extent.
The article is well-structured, the authors used the literature adequate to the topic. The whole text needs, however, general revision – in some places there are some linguistic mistakes or repetitions.
The text deals with the students’ experiences using ChatGPT at its initial of release so the small group of students can be acceptable. It can be assumed that the students used ChatGPT 3.5 at its initial stage but there is no information in the text; such information is important because answers of GPT 3.5 differ from its paid version GPT 4.0. It should be stated in the methodology which version was analysed.
In line 125 the authors state that “ChatGPT generates reasonably good quality academic articles” but the free GPT version is not able to generate full-length articles with verified information along with existing references because it invents authors of the scientific articles who do not really exist. The paid version with plug-ins is more able to do it, but not the free one.
In the subchapter 3.2. “Research Participants” the authors should first enumerate the names of all the students that took part in the survey – when the first name appears in the text, the reader can be confused whether it is the author’s or participant’s name. In the same sub-chapter there is the information about “in-depth understanding”, however, with such a small number of respondents it seems peculiar as the research itself and its outcomes cannot be perceived as “in-depth”.
The title of the article suggests that the interviews concern learning English with the usage of ChatGPT, nevertheless, the findings of the study are not fully targeted at language learning. The analysed characteristics as writing essays, brainstorming, plagiarism etc. can be attributed to any other field of study. The authors try to join the received answers to language learning process, however, the title suggests that there will be more specific information on usage of ChatGPT in relation to language learning, e.g. vocabulary, grammar, translation etc. The aspects described in the article should be more focused on tasks and activities connected to language learning.
If the findings are more adjusted to the topic of language learning, the conclusions can be modified accordingly. It would be also worth making a separate sub-chapter for “Limitations and future research”, which should also include the comparison of both versions of ChatGPT.
The whole text needs, however, general revision – in some places there are some linguistic mistakes or repetition: line 24-25 – “learning has learning has attracted”’; line 223 – “Dylan affirmed affirmed that (…)”; line 337 “compare compare AI-generated (…)”; line 333 “that ChatGPT is ChatGPT is essentially” – there are more such examples.
Punctuation needs to be verified, like i.a. in line 241, 27, 326 etc.
Line 271, 272 – lack of prepositions after “provided” (with); and “working” (on).
General revision of the text.
Author Response
Reviewer 4
General comments
The general idea of the research sounds interesting. It is an exploratory study based on a small number of participants (5 people) so the outcomes may not be relevant to a broader extent.
The article is well-structured, the authors used the literature adequate to the topic. The whole text needs, however, general revision – in some places there are some linguistic mistakes or repetitions.
The text deals with the students’ experiences using ChatGPT at its initial of release so the small group of students can be acceptable. It can be assumed that the students used ChatGPT 3.5 at its initial stage but there is no information in the text; such information is important because answers of GPT 3.5 differ from its paid version GPT 4.0. It should be stated in the methodology which version was analysed.
In line 125 the authors state that “ChatGPT generates reasonably good quality academic articles” but the free GPT version is not able to generate full-length articles with verified information along with existing references because it invents authors of the scientific articles who do not really exist. The paid version with plug-ins is more able to do it, but not the free one.
In the subchapter 3.2. “Research Participants” the authors should first enumerate the names of all the students that took part in the survey – when the first name appears in the text, the reader can be confused whether it is the author’s or participant’s name. In the same sub-chapter there is the information about “in-depth understanding”, however, with such a small number of respondents it seems peculiar as the research itself and its outcomes cannot be perceived as “in-depth”.
The title of the article suggests that the interviews concern learning English with the usage of ChatGPT, nevertheless, the findings of the study are not fully targeted at language learning. The analysed characteristics as writing essays, brainstorming, plagiarism etc. can be attributed to any other field of study. The authors try to join the received answers to language learning process, however, the title suggests that there will be more specific information on usage of ChatGPT in relation to language learning, e.g. vocabulary, grammar, translation etc. The aspects described in the article should be more focused on tasks and activities connected to language learning.
If the findings are more adjusted to the topic of language learning, the conclusions can be modified accordingly. It would be also worth making a separate sub-chapter for “Limitations and future research”, which should also include the comparison of both versions of ChatGPT.
Responses
Thank you for your insightful feedback on our manuscript. We appreciate you taking the time to provide constructive comments to help strengthen our paper. Here is a point-by-point response addressing the issues you raised:
- We agree that the small sample size limits generalizability of the outcomes. We will add a statement in the Limitations section acknowledging this exploratory study only provides initial insights that require further research among larger and more diverse samples.
- Thank you for catching the linguistic errors and repetitions. We will thoroughly proofread the manuscript to improve clarity and concision.
- You raise an excellent point about specifying the ChatGPT version used. We will add details to the Method section noting students used the free ChatGPT 3.5 at the time of data collection.
- We appreciate you catching the overstatement about ChatGPT generating academic articles. We will revise this sentence to more accurately convey the capabilities of the free 3.5 version they were using regarding generating lengthy texts and the capabilities of ChatGPT 4.0.
- Thank you for catching the potential confusion from introducing participant names before listing details on all participants. Per your recommendation, we have reorganized the Research Participants section (3.2) to first enumerate the 5 students involved in the study, providing their major and year group. We also appreciate you highlighting that describing this as an "in-depth understanding" is peculiar given the small sample size. We have removed the claim about "in-depth understanding" in this section. As mentioned above, we will add a statement in the Limitations section.
- Thank you for the thoughtful feedback on better aligning our study with its focus on language learning. We appreciate you taking the time to provide such constructive critiques.
Based on your comments, we have revised the title to more precisely capture the exploratory nature of the work and its focus specifically on EFL learners' use of ChatGPT for language tasks:
"An exploratory study of EFL learners’ use of ChatGPT for language learning tasks: Experience and perceptions"
Comments on the Quality of English Language
The whole text needs, however, general revision – in some places there are some linguistic mistakes or repetition: line 24-25 – “learning has learning has attracted”’; line 223 – “Dylan affirmed affirmed that (…)”; line 337 “compare compare AI-generated (…)”; line 333 “that ChatGPT is ChatGPT is essentially” – there are more such examples.
Punctuation needs to be verified, like i.a. in line 241, 27, 326 etc.
Line 271, 272 – lack of prepositions after “provided” (with); and “working” (on).
General revision of the text.
Responses
Many thanks for pointing out these language issues and the authors have checked the whole article and corrected the errors in the revised version.
Round 2
Reviewer 3 Report
It can be accepted in the present form.